# Physical Activity and Low Glycemic Index Mediterranean Diet: Main and Modification Effects on NAFLD Score. Results from a Randomized Clinical Trial

**DOI:** 10.3390/nu13010066

**Published:** 2020-12-28

**Authors:** Isabella Franco, Antonella Bianco, Antonella Mirizzi, Angelo Campanella, Caterina Bonfiglio, Paolo Sorino, Maria Notarnicola, Valeria Tutino, Raffaele Cozzolongo, Vito Giannuzzi, Laura R. Aballay, Claudia Buongiorno, Irene Bruno, Alberto R. Osella

**Affiliations:** 1Laboratory of Epidemiology and Biostatistics, National Institute of Gastroenterology, “S. de Bellis” Research Hospital, Castellana Grotte (Ba), Via Turi 27, 70013 Castellana Grotte, Italy; isabella.franco@irccsdebellis.it (I.F.); antonella.bianco@irccsdebellis.it (A.B.); antonella.mirizzi@irccsdebellis.it (A.M.); angelocampanella7@gmail.com (A.C.); catia.bonfiglio@irccsdebellis.it (C.B.); paolosorino96@libero.it (P.S.); buongiorno.claudia@gmail.com (C.B.); irenebrunodiet@gmail.com (I.B.); 2Laboratory of Nutritional Biochemistry, National Institute of Gastroenterology, “S. de Bellis” Research Hospital, Castellana Grotte (Ba), Via Turi 27, 70013 Castellana Grotte, Italy; maria.notarnicola@irccsdebellis.it (M.N.); valeria.tutino@irccsdebellis.it (V.T.); 3Department of Gastroenterology, National Institute of Gastroenterology, “S. de Bellis” Research Hospital, Castellana Grotte (Ba), Via Turi 27, 70013 Castellana Grotte, Italy; raffaelecln1@alice.it (R.C.); v55567@alice.it (V.G.); 4Human Nutrition Research Center (CenINH), School of Nutrition, Faculty of Medical Sciences, Universidad Nacional de Córdoba, Enrique Barros Pabellón Biología Celular, Ciudad Universitaria, Cordoba X5000, Argentina; laballay@fcm.unc.edu.ar

**Keywords:** physical activity, non-alcoholic fatty liver disease, diet, controlled attenuation parameter (CAP), aerobic exercise, combined exercise

## Abstract

Background: Non-Alcoholic Fatty Liver Disease (NAFLD) is the most common chronic liver disease worldwide, and lifestyle modification is the current standard treatment. The aim of the study was to estimate the effect of two different physical activity (PA) programs, a Low Glycemic Index Mediterranean Diet (LGIMD), and their combined effect on the NAFLD score as measured by FibroScan^®^. Methods: Moderate or severe NAFLD subjects (*n* = 144) were randomly assigned to six intervention arms during three months. Interventions arms were a control diet, LGIMD, aerobic activity program (PA1), combined activity program (PA2), and LGIMD plus PA1 or LGIMD plus PA2. The data were compared at baseline, at 45 days, and at 90 days. Analysis of variance was performed under the intention-to-treat principle. Results: There was a statistically significant reduction in the NAFLD score after 45 days of treatment in every working arm except for Arm 1 (control diet). After 90 days, the best results were shown by the intervention arms in which LGIMD was associated with PA: LGIMD plus PA1 (−61.56 95% CI −89.61, −33.50) and LGIMD plus PA2 (−38.15 95% CI −64.53, −11.77). Conclusion: All treatments were effective to reduce NAFLD scores, but LGIMD plus PA1 was the most efficient.

## 1. Introduction

Non-Alcoholic Fatty Liver Disease (NAFLD) is a broad spectrum of liver diseases. It is characterized by fat accumulation in the liver in the absence of excessive alcohol consumption or any other specific causes of hepatic steatosis [1]. NAFLD is a major public health challenge as it is associated with the rising prevalence of obesity and Diabetes Mellitus type 2 worldwide and is a risk factor for Cardiovascular Diseases. Moreover, NAFLD is the most common cause of chronic liver disease and increases the 5-year direct and indirect health care costs by an estimated 26% [2]. The global prevalence of NAFLD is currently estimated to be 24% and is continually rising [3]. Our studies, conducted in the population of Castellana Grotte and Putignano (district of Bari, Apulia, Italy), evidenced an NAFLD prevalence of about 30%, with a higher prevalence in male and older patients [4,5]. Therefore, accurate diagnosis of NAFLD is important in patient management [6]. Until now, the gold standard for hepatic steatosis diagnosis has been liver biopsy, but it is an invasive procedure that has several disadvantages such as low acceptance for patients (especially for volunteers), complications, and even a small risk of death (0.01%) [7,8]. Today, many imaging methods are available for the non-invasive assessment of NAFLD [9], but they have limitations such as high costs or operator-dependency of Ultrasound Scan (US) [10,11]. FibroScan^®^ is a non-invasive examination that appears to have good repeatability, which can be used as indices of continuous observation [12]. Data suggest that the controlled attenuation parameter (CAP), measured with FibroScan^®^, can be compared to liver biopsy for the detection and quantification of steatosis [13], and recent studies have evaluated its accuracy in NAFLD [14]. Furthermore, Lee et al. 2016 demonstrated that CAP and liver stiffness can be used as non-invasive and predictive markers of steatosis and fibrosis in patients with NAFLD [15].

Lifestyle modification is the standard treatment for NAFLD [16]. This approach encompasses dietary modifications, exercise, reduced alcohol intake, and weight loss, as they offer a range of health benefits. The Mediterranean Diet (MD) has long been associated with favorable health outcomes [17,18], and good adherence to MD has been reported to have a beneficial effect on the severity of NAFLD [19] as well as of Cardiovascular Diseases [20]. In particular, the Low Carbohydrate Diet, similar to the Low Glycemic Index Mediterranean Diet, is characterized by a reduction in the intake of several ultra-processed foods, refined grains starches, and foods rich in simple or added sugars, and it leads to a number of benefits such as weight loss and weight loss maintenance, reduction of Diastolic Blood Pressure (DBP), reduction of low-density lipoprotein cholesterol (LDL-C) and Triglycerides levels, increase of high-density lipoprotein cholesterol (HDL-C) levels, improvements in insulin resistance, and reduction of glycated hemoglobin (HbA1c) levels [21].

In addition, different forms of exercise have been shown to have similar effects on liver fat [22,23] if physical activity (PA) guidelines recommendations are followed [24]. Minimal physical activity (PA) decreases by 6% the risk for NAFLD compared to sedentary behavior [25]. The European Association for the Study of the Liver (EASL), the European Association for the Study of Diabetes (EASD) and the European Association for the Study of Obesity (EASO) guidelines recommend 150 to 200 min/week of moderate-intensity aerobic physical activity in three to five sessions [26]. Aerobic activity and resistance training seem to have similar improvement in NAFLD but with different mechanisms [27]. Nevertheless, there are still no universal recommendations on the optimal intensity, dose, and type of physical activity in NAFLD treatment. Two studies conducted in this area have confirmed the efficacy of Low Glycemic Index Mediterranean Diet (LGIMD) [28] and the efficiency of aerobic PA on NAFLD [29]. In this context, and as studies about the effect of diet and PA on NAFLD are scarce, we hypothesized that different PA programs and LGIMD or their combination could have different degrees of efficacy on NAFLD severity. We built a local version of the Mediterranean diet based on the work of Trichopoulos et al. and Elia, which contains no more than 10% of total daily calories deriving from saturated fats and a low glycemic index with an extensive use of olive oil [30,31]. Furthermore, the PA programs were built following European guidelines, which comprise aerobic PA as well as resistance training with precision indication about frequency, duration and intensity [26]. This work was aimed estimate the effect of healthy diet (LGIMD), two different PA programs, and their combination on the score of NAFLD as measured in a continuous scale by FibroScan^®^. Moreover, we aimed to explore biochemical, anthropometric, and body composition changes by different degrees of severity of NAFLD in geographical area where the Mediterranean diet is the most prevalent way of eating.

## 2. Materials and Methods

### 2.1. Participants

Details about the study have been published elsewhere [32]. Briefly, 166 subjects, referred to the Laboratory of Epidemiology and Biostatistics of the National Institute of Digestive Diseases, IRCCS “S. de Bellis”, Castellana Grotte, Italy, by their General Practitioners or identified during the NutriEp [5] or MICOL [33] enrollment or follow-up process, were assessed for eligibility. The trial was conducted from March 2015 to December 2016. Patients were consecutively enrolled, and the trial was registered at www.https://clinicaltrials.gov (registration number CT02347696). All subjects provided informed consent and the study was conducted in accordance with the Helsinki Declaration and approved by the Ethical Committee (Prot. n. 10/CE/De Bellis, 3 February 2015).

### 2.2. Study Design

This study was a parallel group randomized controlled clinical trial. Inclusion criteria were Body Mass Index (BMI) ≥25, expressed as weight in kilograms divided by square height in meters (kg/m^2^); age >30 years old and <60 years old; moderate or severe NAFLD as assessed by the controlled attenuation parameter (CAP). Exclusion criteria included (1) overt cardiovascular disease and revascularization procedures; (2) stroke; (3) clinical peripheral artery disease; (4) T2DM (current treatment with insulin or oral hypoglycemic drugs, fasting glucose >126 mg/dL, or casual glucose >200 mg/dL; (5) more than 20 gr/daily of alcohol intake; (6) any severe medical condition that could prevent the subject’s participation in a nutritional intervention study; (7) subjects following a special diet or involved in a program for weight loss, or who had experienced recent weight loss, and (8) inability to follow a LGIMD for religious or other reasons [32].

### 2.3. Sample Size

Sample size was estimated considering the repeated measurements nature of the outcome. From a previous study [34], the mean ± Standard Deviation (SD) score of NAFLD was estimated to be 4.5 (1) and 4.0 (0.5) for the treatment and control groups, respectively. We hypothesized a decrease by 20 units of the CAP score after three months of intervention. Probabilistic type I error was fixed at 0.05 (one sided) and probabilistic error type II was fixed at 0.10; statistical power was fixed at 0.9. The correlation between baseline/follow-up measurements of the outcome was set at 0.4. A sample size of *n*1 = *n*2 = *n*3 = *n*4 = *n*5 = *n*6 = 20 was estimated.

### 2.4. Data Collection

During enrollment, patients were interviewed by trained nutritionists to collect information on socio-demographic aspects, medical history, and lifestyle. Information about physical activity was collected using the validated questionnaire International Physical Activity Questionnaire Long Form (IPAQ-LF) [35]. The European Prospective Investigation into Cancer and Nutrition Food Frequency Questionnaire (EPIC FFQ) was used to probe alcohol intake and eating behavior [36]. Anthropometric measurements (weight, height, waist circumference) were taken in a standard manner. Weight and height measurements were taken using SECA instruments (Model 700; and Model 206; 220 cm; SECA, Hamburg, Germany). Blood samples, taken by venous puncture, were drawn in the morning after overnight fasting and were collected in tubes containing ethylenediamine tetra acetic acid (K-EDTA) anticoagulant. Biochemical measurements were performed using standard methods.

### 2.5. Randomization and Masking

According to a computerized random numbers sequence, subjects were randomized to six arms as follows: (1) Control Diet (CD) based on CREA-AN (Research Center for Food and Nutrition, Council for Agricultural Research and Economics, Rome, Italy) guidelines [37]; (2) Low Glycemic Index Mediterranean Diet (LGIMD); (3) Physical Activity aerobic program (PA1); (4) Physical Activity combined program (aerobic activity and resistance training) (PA2); (5) LGIMD plus PA1 and (6) LGIMD plus PA2. The trial flowchart is shown in Figure 1.

### 2.6. Dietary Interventions

Two types of diets were prescribed: a Control Diet (CD) based on CREA-AN guidelines [37] and a Low Glycemic Index Mediterranean Diet (LGIMD) [28]. No indications were given regarding the total calories to be consumed. Foods in LGIMD have all a low Glycemic Index (GI) and no more than 10% of total daily calories coming from saturated fats. The LGIMD was high in monounsaturated fatty acids (MUFA) from olive oil and contained also omega-3 polyunsaturated fatty acids (ω3PUFA), from both plant and marine sources. The prescribed diets were provided in brochure format, with graphic explanations. All participants were asked to record what they ate on a daily diary. The Mediterranean Adequacy Index (MAI) was chosen as a relevant measure to evaluate the adherence to both the intervention diet and the control diet [38]. Detailed information relating to the LGIMD intervention and control (recommended by the WHO and followed by INRAN [39]) is shown in Appendix A and Appendix A.

### 2.7. Physical Activity Interventions

To evaluate the initial physical condition, the right training program, and to compare the initial with the FU assessment of physical condition, three field tests were carried out: cardiorespiratory [40], strength [41], and flexibility fitness [42]. Cardio-respiratory fitness was assessed by means of the 2 km walking test, which is suitable for adults [40], whereas strength and flexibility fitness were evaluated by means of the push-up test (also called press-up test) and the Sit and Reach test, respectively. Subjects randomized to an intervention arm that included physical activity underwent the tests at baseline, after 45, and after 90 days, whereas subjects randomized to the diet arms underwent the field fitness tests at baseline and after 90 days. Physical activity interventions included two different types of exercise programs: Physical Activity 1 (PA1), based on the aerobic activity program, and Physical Activity 2 (PA2), based on the combination of aerobic activity and resistance training. The primary methods used for determining the intensity of prescribed exercise are maximum heart rate (Max HR), through heart-rate monitors [43], and metabolic equivalent (MET) [44,45]. To establish the age-predicted maximal heart rate, we used the formulas of Tanaka [46]. All subjects were randomized for one of the two physical activity programs based on the results of the tests performed. A progressive change of the monthly target has been planned based on the results obtained.

#### 2.7.1. Aerobic Activity Program

Three non-consecutive sessions per week of moderate intensity aerobic activity (60–75% max HR, 3.0–5.9 METs distributed as follows: week 1–4: 14.2 kcal ∗ kg^−1^ ∗ week^−1^; week 5–8: 18.9 kcal ∗ kg^−1^ ∗ week^−1^; week 9–12: 23.6 kcal ∗ kg^−1^ ∗ week^−1^). The approximate duration of each session was between 50 and 60 min. Aerobic exercise intensity was monitored every 5 min using an automated heart rate monitor. Total weekly exercise duration was 150/180 min. Exercise modalities included treadmill walking, cycling, cross-training, and rowing.

#### 2.7.2. Combination of Aerobic Exercise and Resistance Training

Three non-consecutive sessions including (1) 45 min of moderate intensity aerobic exercise on treadmill, cycling, rowing, and cross-trainer (60–75% max HR, 3.0–5.9 METs distributed as follows: week 1–4: 14.2 kcal ∗ kg^−1^ ∗ week^−1^; week 5–8: 18.9 kcal ∗ kg^−1^ ∗ week^−1^; week 9–12: 23.6 kcal ∗ kg^−1^ ∗ week^−1^); (2) 3 sets of 12 exercises, each to volitional fatigue like leg press, adductor/abductor machine, gluteus machine, bicep curls, triceps extension, three different abdominal exercises, leg machine, low row, shoulder flexion. The weight lifting was increased by 1–2.5 kg ∗ week^−1^ when 10 repetitions were completed in good form. Total weekly exercise duration was 180/240 min. All physical activity programs were carried out in a local gym, and all subjects participated in the physical training program, deciding on the times and days for the training, and interacting with the specialist at any time.

### 2.8. Outcome Assessment

To quantify and detect hepatic steatosis, the controlled attenuation parameter (CAP) score was used. CAP measures the degree of ultrasound attenuation due to hepatic fat at the standardized frequency of 3.5 MHz through a vibration-controlled elastography (VCTE) implemented on FibroScan^®^ (Echosens, Paris, France). Values <248, 248–267, 268–279 and ≥280 dB/m indicate absence, mild, moderate, or severe NAFLD, respectively [47].

### 2.9. Statistical Analysis

The intention-to-treat principle was applied in all analysis. A description of the data was performed by Means (±SD), Median (IQR), and Frequencies (%) as appropriate. Data were also described separately by NAFLD severity. Biochemical markers were included only for descriptive purposes. Adherence to PA programs was evaluated as the proportion of effective frequency/time/intensity spent on gym or field sessions divided by the expected weekly frequency/time/intensity per session for the aerobic exercise group, plus expected load for the combined exercise group. The adherence was estimated by age group, gender, and month, and it was expressed as percentage. The third to tenth weeks of intervention were considered. The first two weeks were excluded to give subjects the opportunity to adapt to the programs training, and the last two weeks were excluded because the compliance could be decreased. The adherence to the LGIMD was measured with MAI [38]. Random week and weekend days were chosen from the second to 9th weeks of intervention. The first two weeks were excluded to allow participants to learn about the new diet and weeks successive to ninth were excluded because the compliance could decrease. To clearly describe the adherence and compliance to the LGIMD, MAI was estimated by gender, age, and week. Compliance was defined as previously described [38]. A median value of 7.5 was expected, as established by the reference Italian Mediterranean Diet [38].

To analyze differences among groups (Diet by Time), an Analysis of Variance for repeated measures was applied to the data. This statistical technique was chosen because it is relatively robust against violations of its assumptions, allows for multiple comparisons, and can be applied to a variety of experimental designs. Analysis of variance for repeated measurements was also performed by NAFLD severity. CAP was normally distributed, so no transformation was needed. Post estimation procedures were performed to compare estimated means with the reference category (control diet at baseline). To display graphically prediction from the fitted model, margins post estimation commands were performed. Margins statistics were calculated from predictions of a previously fitted Analysis of Variance model and displayed in graph form. Results are all expressed in the natural scale of CAP measurements as long as 95% Confidence Interval (95% CI).

Stata statistical (version 16.1) package was used to perform statistical analysis (StataCorp, 4905 Lakeway Drive, College Station, TX, USA); the Stata code is available upon request from the email: arosella@irccsdebellis.it).

## 3. Results

One hundred and sixty-six subjects were assessed for eligibility: 17 subjects were excluded because they did not satisfy the inclusion criteria and 5 were excluded for various other reasons. The study design is shown in Figure 1. The following subjects were lost to follow-up: 5, 3, 7, 6, 9, and 4 from Arms 1, 2, 3, 4, 5 and 6, respectively.

Finally, 144 subjects were included and randomized to the intervention arms. Characteristics of the participants are shown in Table 1 and Table 2 and in Appendix B, Table A1.

As expected, the age–sex distribution of the population under study reflected the age–sex distribution of the NAFLD condition in the population [4]. About 62% of the sample were men, and the mean age was 49.9 years (10.08), but few subjects were under 40 years old (12.5%). As expected, all subjects were overweight or obese with increased waist circumference. At baseline, almost every participant had low levels of HDL-C and a high HOMA-IR. Most subjects were married and had attended secondary school or higher (Table 2). All parameters considered were equally distributed among groups at baseline with the exception of NAFLD severity. More subjects had severe than moderate NAFLD, and the CAP value was over 323 dB/m in all groups with a successive decreasing in intermediate and final observation times.

The behavior of anthropometric, body composition, and biochemical profiles by time and intervention arm are shown in Appendix B, Table A1. Overall, there were improvements in all markers, especially at 45 days. These improvements were particularly evident for intervention arms PA1 + LGIMD and PA2 + LGIMD.

Field Fitness Test results by gender and time are shown in Table 3.

Overall, there was a significant improvement of the fitness index in both men and women. The compliance to physical activity programs is shown in Table 4.

Overall, adherence was above 80% to both PA intervention program in both genders and in all weeks.

Compliance to LGIMD, as measured by MAI, by age class, gender, and week is shown in Table 5.

Compliance with the control diet showed a median MAI of 1.29 (0.77, 2.63). MAI was lower amongst males (1.07) than females (2.14), whereas compliance with LGMID evidenced a median MAI of 11.2 (6.4, 22.8). Adherence was lower among women (10.0) than men (11.3).

Table 6 shows the changes in the CAP value by intervention arms and time (at 45 days and at 90 days).

Apart from the CD, there were statistically significant principal effects of all the other intervention arms, Arms 3 and 5 showing the strongest effects (PA1: −166.35, 95%CI −242.01; −90.68 and LGIMD plus PA1: −94.10, 95% CI −170.87; −19.12). After the analysis, no statistical significance principal effect of time in reducing CAP was observed. All comparisons between different time points and Intervention Arms, except for Intervention Arm 1, showed a statistically significant decreased value for CAP. The strongest effect was observed for LGIMD plus PA1 at 45 days (−52.96, 95% CI −79.03; −26.88) and 90 days (−61.56, 95% CI −89.61; −33.50).

Results from separate analysis by NAFLD severity are shown in Appendix B, Table A2 and Table A3. The relative decrease of CAP values was more relevant among severe NAFLD subjects. HOMA-IR decreased in all intervention arms with the exception of LGIMD in moderate NAFLD subjects. In both moderate and severe NAFLD subjects, there was a modest effect of only-diet intervention arms on CAP. Only-PA intervention arms effect on CAP values evidenced a decrease among severe NAFLD subjects but not among moderate ones. The most intense effect in decreasing CAP values was observed among severe NAFLD subjects who underwent PA1+LGIMD. An improvement in all anthropometric, body composition, and biochemical parameters was observed (most evident at day 45). Noteworthy, the improvement in HDL-C was accompanied by an improvement in FFM only in moderate NAFLD subjects. Table 7 shows the changes in the CAP value by NAFLD severity, intervention arms, and time (at 45 days and at 90 days).

There was an effect that was statistically significant only in subjects with moderate NAFLD allocated to Arm 6 after 90 days, whereas among severe NAFLD subjects, there were statistically significant effects in all times and arms with the exception of CD and Arm 6 at 90 days. The strongest estimated effect was observed in the PA1 + LGIMD intervention arm.

The results of predictive margins from analysis of variance are graphically displayed in Figure 2, which highlights the strong effect of LGIMD plus PA1. Hypothesis testing about the equality of measurements at baseline showed no statistically significant differences in CAP among intervention arms (*p* > 0.05).

## 4. Discussion

In this RCT, the combination of the aerobic activity program and an LGIMD was associated with the strongest reduction of the CAP, as measured by FibroScan^®^, compared to the other intervention arms. Overall, this effect was greater until 45 days of intervention. However, the effect of Aerobic PA plus LGIMD was stronger during the complete intervention time in severe than moderate NAFLD subjects. The effect on CAP was accompanied by an overall improvement in all anthropometric, body composition and biochemical parameters.

Currently, a lifestyle change focused on diet and physical activity seems to be the best therapy in the treatment of NAFLD, combined with pharmacologic therapy when necessary [48]. Regular physical activity is now considered a therapy to prevent the onset and progression of several chronic diseases, including NAFLD [49]. Aerobic, resistance and combination exercise programs may enhance systemic indicators of hepatic functions and intrahepatic fats in NAFLD patients in both mild and advanced cases [50].

The adoption of either aerobic or strength PA programs showed to influence hepatic metabolism with a resulting decrease in hepatic fat accumulation, an increased insulin sensitivity, and fat oxidation. Moreover, these effects have been observed even without weight loss [51].

Although the risk of developing NAFLD or more advanced grades of liver diseases seems to be exercise dose-related [52], no differences among different intensity aerobic exercise regimens were recently found: all of them reduced liver fat [53].

At the community level, a reduced infiltration of liver fat has been documented after a combined exercise program [54]. This effect was proportional to weight loss rate and event without it.

LGIMD was locally created by integrating the works of Trichopolou et al. [30] and Elia [31]. Moreover, we have adapted it to our population, and it is similar to the diet suggested by other authors for the metabolic syndrome [55]. LGIMD, which contains no more than 10% of total daily calories deriving from saturated fats and a low glycemic index, was associated with a more intense reduction of the NAFLD score [28]. The LGIMD is rich in monounsaturated fatty acids (MUFA) and also contains omega-3 polyunsaturated fatty acids (ω3PUFA).

In this RCT, in working Arm 2, where the subjects only followed a LGIMD, we observed a statistically significant improvement of the score of NAFLD, reducing the CAP value, especially at 45 days. However, after 45 days, a slight increase in the CAP value was observed. Several studies have shown that lifestyle changes are difficult to implement and maintain for a long time, although individuals know the importance of having a healthier lifestyle [56,57]. In our study, adherence was almost homogenously distributed during the intervention time but, as it has been shown, participants could have given answers that are socially desirable [58].

When comparing the effectiveness of the two different physical activity programs, it is clear that both the aerobic and the combined exercise program significantly reduced the CAP value. Our data are confirmed by previous studies that have demonstrated the effectiveness of aerobic activity on liver function [59,60]. Moreover, recently, resistance training has also been found to be effective in NAFLD treatment, regardless of weight loss [61,62,63].

Both aerobic exercise and combined exercise significantly reduced the CAP value from 45 days onward until the end of the project. The literature exhaustively showed the efficacy of both combined and resistance training in NAFLD [64]. Furthermore, aerobic activity and resistance training have been recently shown, although with different mechanisms, to improve NAFLD score [23,65]. At least three mechanisms have been proposed for aerobic PA: activation of lipolysis in different tissues, upregulation of the uncoupling protein-1 and peroxisome proliferator-activated receptor γ pathways, and alteration of adipokine levels [23]; on the other hand, resistance training stimulates the hypertrophy of type II muscle fibers, alters myokine levels, and activates glucose transporter 4, AMP-activated protein kinase, and caveolins [23]. Moreover, resistance training appeared to improve NAFLD with less energy consumption. A recent study conducted in mice with diethyl nitrosamine-induced hepatocellular carcinoma has shown that in early stages, voluntary exercise decreases the proliferation rate of dysplastic hepatocytes [66].

Therefore, PA could be hypothesized as a preventive measure for liver cancer development, as NAFLD is a stage of a broad spectrum of conditions that lead to hepatocellular carcinoma. In this sense, after an energy restriction dietary intervention and/or regular PA, several facts have been observed such as histological improvements, resolution of liver fat, as well as necroinflammation and fibrosis. These changes have been observed even after 7% to 10% of weight loss [67,68].

Healthy dietary habits and healthy lifestyle, similar to any regular physical activity, can improve NAFLD more than weight loss alone [69].

Weight loss achieved by dietary treatment must be done taking into account quantitative and qualitative characteristics, as energy restriction per se is not sufficient to improve NAFLD [22]. Moreover, a dietary composition modulating both macro and micronutrients is crucial [70]. Therefore, balanced nutrition such as LGIMD, moderate weight loss, and a physical activity program can be considered as the best therapeutic approach in NAFLD.

This seems to be confirmed in our study where the effect of intervention was more evident in the severe NAFLD group, highlighting that the combination of diet and physical activity can be an effective therapeutic tool in the prevention not only of the complications of NAFLD but also of the associated cardiovascular risk. The observed effect in the severe NAFLD group could be due to the particular distribution of NAFLD severity, as 75% of participants (data not shown) had severe NAFLD at baseline. Moreover, the use of FibroScan^®^ as the NAFLD assessment method could have enhanced the frequency of severe NAFLD diagnosis. It is known that FibroScan^®^ has a greater sensibility [71] than Liver Ultrasound for fat infiltration >25–30% of hepatocytes [72].

The diminution of intrahepatic fat content is not the only marker of interest in this study. The metabolic pattern resulting from the interventions was explored, and the results are almost always in line with the literature. The negative association between adherence to Mediterranean diet and NAFLD severity has been confirmed in this study as well as HOMA-IR decreasing, which has been more strong in severe NAFLD subjects [19]. As noted above, we built a local version of the Mediterranean diet following the European Association for the study of the Liver, the European Association for the study of Diabetes, and the European Association for the study of Obesity guidelines [26]. After the intervention period, our results are similar to those of Ryan et al., who found an improvement in fatty liver content and HOMA-IR [73]. Moreover, a recent systematic review and meta-analysis has documented beneficial changes of the Mediterranean diet on metabolic health, including anthropometric and biochemical markers [74].

As regards PA, a network meta-analysis has shown that aerobic exercise plus diet and aerobic exercise alone exert an improvement on BMI and HOMA-IR but progressive resistance training has an effect only on HOMA-IR but not on BMI [75]. While diet seems to be more effective at improving intrahepatic markers of liver damage, PA plus diet improves insulin resistance and BMI [76]. However, in our study, there was a modest effect of only the LGIMD intervention arm. In the geographical area where the study was conducted, the Mediterranean diet is the most prevalent dietary pattern with a high adherence at population level, which may be the cause of the modest observed effect [76].

This study has several strengths and limitations. Strengths include the study design, the measured compliance to both dietary and PA interventions as well as their controlled application, and an adequate sample size. Moreover, a well-validated assessment of the outcome such as FibroScan^®^ has been implemented. The applied intention-to-treat analytical strategy prevents the design from introducing bias relating to non-adherence to the protocol to the prognosis; therefore, this RCT provides an unbiased assessment of treatment efficacy [77]. It is worthy to note that in this area, the most prevalent dietary pattern is the local version of Mediterranean Diet; then, a dilution bias could be present. Another limitation may be the duration of the intervention, which prevents a wide application in the clinical field, However, the objective of the study was to estimate the effect of the intervention in order to establish its efficacy.

## 5. Conclusions

It is clear that PA single (aerobic or resistance exercise) or combined (aerobic plus resistance exercise) should be integrated into NAFLD management [78]. Moreover, aerobic exercise programs are simple to implement and the cost-efficiency is higher. These should be included in programs for the primary prevention of metabolic and cardiovascular diseases.

In conclusion, a multidisciplinary team including dietitians, a psychologist, and physical activity supervisors is needed to ensure the best management of NAFLD patients [78].

## Figures and Tables

**Figure 1 nutrients-13-00066-f001:**
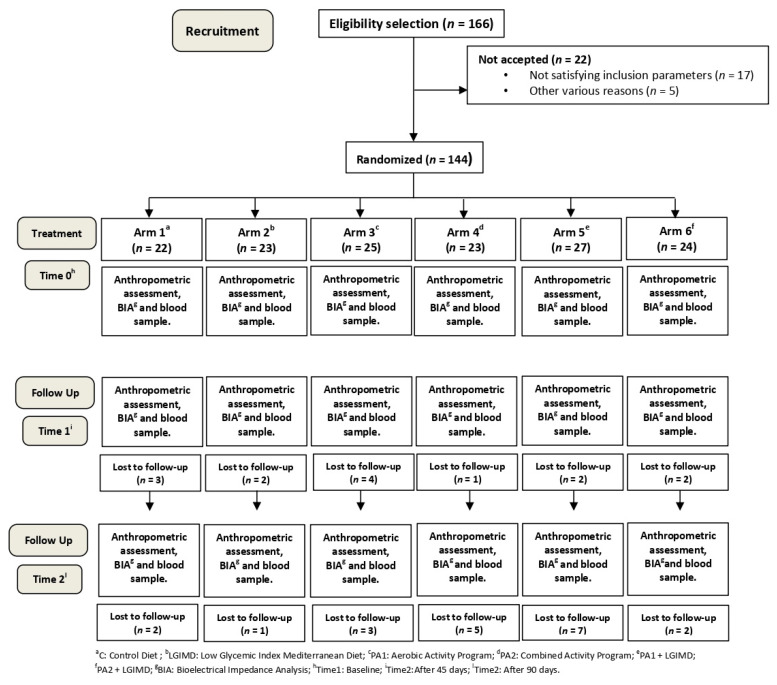
Flowchart of the study.

**Figure 2 nutrients-13-00066-f002:**
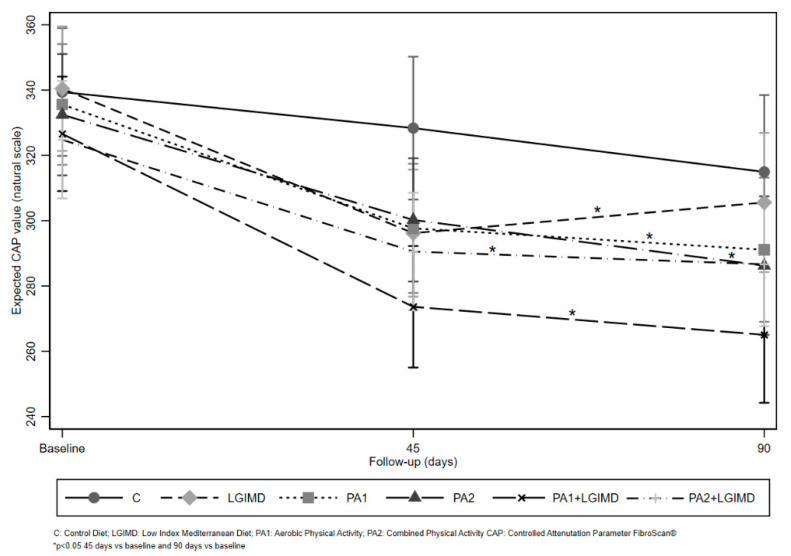
Diet and Physical Activity Effects on NAFLD: Expected CAP values by Treatment and Time.

**Table 1 nutrients-13-00066-t001:** Characteristics of Participants by Intervention Arms and Time. NutriAtt Trial, Castellana Grotte, 2015–2016.

Variables	Time	Working Arms						
		C ^a^	LGIMD ^b^	PA1 ^c^	PA2 ^d^	PA1 + LGIMD ^e^	PA2 + LGIMD ^f^	*p*-Value
**N**	Baseline	22 (15.3%)	23 (16.0%)	25 (17.4%)	23 (16.0)	27 (18.7%)	24 (16.7%)	
**Age (Years)**		50.70 (8.67)	50.74 (1.75)	50.45 (9.45)	46.23 (9.39)	50.32 (9.61)	46.75 (10.50)	0.11
**Age (Categorical)**	<40	1 (6%)	1 (6%)	1 (6%)	5 (28%)	3 (17%)	7 (39%)	0.025
	40–49	8 (16%)	3 (6%)	10 (20%)	9 (18%)	12 (24%)	8 (16%)	
	≥50	11 (17%)	18 (28%)	9 (14%)	8 (12%)	10 (15%)	9 (14%)	
**Gender**								
	Male	11 (12%)	13 (15%)	14 (16%)	17 (19%)	18 (20%)	16 (18%)	0.57
	Female	11 (20%)	10 (18%)	11 (20%)	6 (11%)	9 (16%)	8 (15%)	
**CAP (dB/m) ^g^**	Baseline	349.09 (60.28)	341.57 (45.17)	339.92 (44.89)	334.78 (51.01)	328.56 (52.05)	323.63 (53.98)	0.58
	45 days	322.12 (83.18)	296.52 (51.10)	295.10 (50.68)	299.55 (63.73)	275.61 (75.84)	289.38 (70.15)	0.39
	90 days	307.80 (73.10)	303.72 (81.70)	287.88 (54.59)	284.11 (62.22)	259.84 (57.01)	289.18 (72.96)	0.21
**NAFLD ^h^**								
**Moderate**	Baseline	6 (13.6%)	5 (11.4%)	6 (13.6%)	7 (15.9%)	10 (22.7%)	10 (22.7%)	0.64
	45 days	3 (7.0%)	8 (19.0%)	7 (17.0%)	10 (24.0%)	5 (12.0%)	9 (21.0%)	0.45
	90 days	4 (13.0%)	3 (9.0%)	3 (9.0%)	8 (25.0%)	5 (16.0%)	9 (28.0%)	0.20
**Severe**	Baseline	16 (16.0%)	18 (18.0%)	19 (19.0%)	16 (16.0%)	17 (17.0%)	14 (14.0%)	0.64
	45 days	13 (21.0%)	10 (16.0%)	12 (19.0%)	10 (16.0%)	10 (16.0%)	8 (13.0%)	0.45
	90 days	12 (18.0%)	11 (17.0%)	15 (23.0%)	8 (12.0%)	10 (15.0%)	9 (14.0%)	0.20

^a^ C: Control Diet; ^b^ LGIMD: Low Glycemic Index Mediterranean Diet; ^c^ PA1: Aerobic Activity Program; ^d^ PA2: Combined Activity Program; ^e^ PA1 + LGIMD; ^f^ PA2 + LGIMD; ^g^ CAP: Controlled Attenuation Parameter measured by FibroScan^®^; ^h^ NAFLD: Non-Alcoholic Fatty Liver Disease.

**Table 2 nutrients-13-00066-t002:** Additional Characteristics of Participants by Intervention Arm and Time. NutriAtt Trial, Castellana Grotte, 2015–2016.

Working Arms							
Variables	C ^a^	LGIMD ^b^	PA1 ^c^	PA2 ^d^	PA1 + LGIMD ^e^	PA2 + LGIMD ^f^	*p*-Value *
**N**	22	23	25	23	27	24	
**Status**							0.32
Single	1 (6%)	1 (6%)	1 (5%)	3 (19%)	2 (10%)	3 (16%)	
Married	17 (94%)	17 (94%)	18 (90%)	11 (69%)	17 (81%)	16 (84%)	
Divorced	0 (0%)	0 (0%)	1 (5%)	0 (0%)	2 (10%)	0 (0%)	
Widowed	0 (0%)	0 (0%)	0 (0%)	2 (13%)	0 (0%)	0 (0%)	
**Study Level**							0.07
Elementary	1 (6%)	2 (11%)	0 (0%)	0 (0%)	1 (5%)	0 (0%)	
Secondary School	10 (56%)	8 (42%)	8 (40%)	3 (18%)	4 (19%)	6 (30%)	
High School	6 (33%)	7 (37%)	9 (45%)	9 (53%)	8 (38%)	12 (60%)	
Diploma University	1 (6%)	1 (5%)	0 (0%)	0 (0%)	0 (0%)	0 (0%)	
University Degree	0 (0%)	1 (5%)	3 (15%)	5 (29%)	8 (38%)	2 (10%)	
**Smoking status**							0.80
Never smoked	13 (68%)	10 (50%)	12 (55%)	10 (56%)	17 (71%)	12 (55%)	
Former smoker	5 (26%)	6 (30%)	5 (23%)	4 (22%)	4 (17%)	4 (18%)	
Current smoker	1 (5%)	4 (20%)	5 (23%)	4 (22%)	3 (13%)	6 (27%)	

^a^ C: Control Diet; ^b^ LGIMD: Low Glycemic Index Mediterranean Diet; ^c^ PA1: Aerobic Activity Program; ^d^ PA2: Combined Activity Program; ^e^ PA1 + LGIMD; ^f^ PA2 + LGIMD; * Fisher exact test.

**Table 3 nutrients-13-00066-t003:** Field Fitness Test Results by Gender and Time. NutriAtt Trial, Castellana Grotte, 2015–2016.

Working Arm		Male			Female		
	Test	Baseline	45 Days	90 Days	Baseline	45 Days	90 Days
		Mean ± SD	Mean ± SD	Mean ± SD	Mean ± SD	Mean ± SD	Mean ± SD
**C ^a^**	2 Walking (sec) *	97.39 (17.50)	N/A ^≠^	103.09 (12.18)	14.33 (56.75)	N/A ^≠^	27.69 (37.69)
	Push-up *	11.75 (8.81)	N/A ^≠^	13.13 (10.05)	4.38 (4.24)	N/A ^≠^	9.50 (5.47)
	Sit and Reach	−11.00 (10.06)	N/A ^≠^	−10.38 (11.11)	−6.25 (10.78)	N/A ^≠^	−9.00 (13.61)
**LGIMD ^b^**	2 Walking (sec) *	105.73 (24.97)	N/A ^≠^	100.05 (12.85)	13.18 (39.20)	N/A ^≠^	39.19 (30.90)
	Push -up *	8.92 (7.80)	N/A ^≠^	11.64 (8.50)	6.38 (5.90)	N/A ^≠^	11.50 (10.01)
	Sit and Reach	−9.00 (9.28)	N/A ^≠^	−11.36 (6.48)	−11.63 (9.50)	N/A ^≠^	−7.00 (8.65)
**PA1 ^c^**	2 Walking (sec) *	87.30 (17.30)	100.81 (16.10)	101.88 (9.15)	27.22 (39.52)	42.60 (21.97)	58.46 (26.18)
	Push-up *	10.50 (7.86)	13.75 (10.93)	15.30 (10.73)	9.13 (7.08)	10.50 (7.52)	15.29 (10.01)
	Sit and Reach	−11.21 (8.72)	−10.50 (9.41)	−7.00 (10.26)	−9.25 (10.87)	−6.25 (9.32)	−4.00 (10.49)
**PA2 ^d^**	2 Walking (sec) *	94.08 (10.74)	96.20 (10.59)	103.83 (12.16)	56.29 (20.92)	63.28 (29.55)	65.18 (36.41)
	Push-up *	11.06 (7.24)	15.19 (6.72)	16.67 (7.79)	6.67 (5.61)	7.33 (5.20)	11.50 (2.89)
	Sit and Reach	−9.81 (6.65)	−7.81 (7.60)	−4.73 (8.55)	−5.83 (6.55)	−5.83 (5.49)	−7.25 (6.08)
**PA1+LGIMD ^e^**	2 Walking (sec) *	82.90 (12.69)	95.85 (11.97)	105.91 (9.26)	33.47 (30.60)	42.09 (25.12)	43.48 (27.77)
	Push -up *	9.06 (6.87)	13.73 (8.56)	17.17 (9.13)	14.56 (11.45)	15.56 (8.75)	15.25 (9.13)
	Sit and Reach	−12.69 (8.10)	−11.53 (8.64)	−9.50 (7.87)	−3.11 (7.77)	−2.67 (7.92)	−2.63 (8.31)
**PA2+LGIMD ^f^**	2 Walking (sec) *	87.26 (16.17)	102.99 (12.36)	102.76 (13.39)	32.24 (26.80)	50.08 (26.30)	61.78 (31.95)
	Push-up *	7.81 (4.86)	11.00 (6.78)	15.14 (8.50)	11.00 (7.52)	16.00 (9.90)	18.25 (7.63)
	Sit and Reach	−18.19 (9.23)	−14.38 (10.31)	−13.79 (11.20)	−4.38 (7.85)	−1.63 (6.97)	0.25 (7.83)

^a^ C: Control Diet; ^b^ LGIMD: Low Glycemic Index Mediterranean Diet; ^c^ PA1: Aerobic Activity Program; ^d^ PA2: Combined Activity Program; ^e^ PA1 + LGIMD; ^f^ PA2 + LGIMD; * Kruskal-Wallis equality-of-populations rank test: *p* < 0.05; ^≠^ N/A = Not Applicable.

**Table 4 nutrients-13-00066-t004:** Adherence to Physical Activity Programs by Working Arm, Gender, and Week from Enrollment. NutriAtt Trial, Castellana Grotte, 2015–2016.

Weeks		PA1 ^a^		PA2 ^b^		LGIMD + PA1 ^c^		LGIMD + PA2 ^d^	
		Male	Female	Male	Female	Male	Female	Male	Female
		Mean (SD)	Mean (SD)	Mean (SD)	Mean (SD)	Mean (SD)	Mean (SD)	Mean (SD)	Mean (SD)
3	Time	96.5 (9.70)	100 (0)	100 (0)	100 (0)	97.8 (8.60)	93.7 (12.40)	100 (0)	100 (0)
	Intensity	101.6 (3.81)	100.5 (1.54)	100.1 (0.61)	100 (0)	99.9 (1.22)	100 (0)	99.9 (0.45)	99.4 (2.41)
	Load	N/A *	N/A *	100 (0)	100 (0)	N/A *	N/A *	100 (0)	100 (0)
4	Time	92.4 (15.8)	89.6 (19.8)	100 (0)	100 (0)	100 (0)	100 (0)	100 (0)	97.4 (9.24)
	Intensity	101.5 (3.20)	100.5 (1.54)	100.1 (0.61)	100 (0)	100.3 (0.63)	100 (0)	99.9 (0.51)	99.4 (2.41)
	Load	N/A *	N/A *	100 (0)	100 (0)	N/A *	N/A *	100 (0)	100 (0)
5	Time	94.4 (12.9)	95.8 (11.8)	95.8 (11.4)	94.4 (13.6)	100 (0)	100 (0)	100 (0)	94.9 (12.5)
	Intensity	101.0 (2.32)	100.1 (0.28)	100.1 (0.61)	100 (0)	100.6 (1.56)	100 (0)	100.1 (0.40)	98.9 (2.81)
	Load	N/A *	N/A *	100 (0)	100 (0)	N/A *	N/A *	100 (0)	100 (0)
6	Time	94.2 (11.5)	100 (0)	93.7 (13.4)	94.4 (13.6)	100 (0)	100 (0)	100 (0)	97.4 (9.24)
	Intensity	100.5 (3.07)	100.1 (0.28)	99.9 (0.41)	100 (0)	100.9 (1.84)	100 (0)	100.1 (0.40)	98.9 (2.81)
	Load	N/A *	N/A *	100 (0)	100 (0)	N/A *	N/A *	100 (0)	100 (0)
7	Time	93.0 (13.2)	92.0 (22.6)	95.8 (11.4)	94.4 (13.6)	97.8 (8.60)	87.5 (24.8)	100.0 (0)	92.3 (20.0)
	Intensity	100.6 (1.53)	100.1(0.28)	100 (0)	100 (0)	100.5 (1.03)	100.0 (0)	100.1 (0.40)	98.9 (2.81)
	Load	N/A *	N/A *	100 (0)	100 (0)	N/A *	N/A *	100 (0)	100 (0)
8	Time	88.2 (19.3)	91.7 (15.4)	95.8 (16.7)	100 (0)	100 (0)	83.3 (27.9)	100 (0)	97.4 (9.24)
	Intensity	100.5 (1.66)	100.1 (0.28)	100 (0)	100 (0)	100.6 (1.08)	100 (0)	100.1 (0.40)	98.9 (2.81)
	Load	N/A *	N/A *	100 (0)	100 (0)	N/A *	N/A *	100 (0)	100 (0)
9	Time	93.4 (16.1)	94.8 (11.7)	93.3 (13.8)	91.7 (16.7)	97.4 (9.24)	88.9 (17.2)	97.9 (8.33)	97.4 (9.24)
	Intensity	99.6 (1.26	100 (0)	100.2 (0.73)	100 (0)	100.1 (1.07)	100 (0)	100.2 (0.79)	98.6 (3.43)
	Load	N/A *	N/A *	100 (0)	100 (0)	N/A *	N/A *	100 (0)	100 (0)
10	Time	98.4 (5.09)	84.5 (19.5)	93.3 (13.8)	91.7 (16.7)	97.4 (9.24)	94.4 (13.6)	97.9 (8.33)	94.8 (12.5)
	Intensity	99.6 (1.26)	100 (0)	100.2 (0.73)	100 (0)	100.1 (1.07)	100 (0)	100.2 (0.79)	98.8 (3.62)
	Load	N/A *	N/A *	100 (0)	100 (0)	N/A *	N/A *	100 (0)	100 (0)

^a^ PA1: Aerobic Activity Program; ^b^ PA2: Combined Activity Program; ^c^ PA1 + LGIMD: Low Glycemic Index Mediterranean Diet; ^d^ PA2+LGIMD: Low Glycemic Index Mediterranean Diet; * N/A = Not Applicable.

**Table 5 nutrients-13-00066-t005:** Mediterranean Adequacy Index by Gender, Age, and Week. NutriAtt Trial, Castellana Grotte, 2015–2016.

			Male			Female	
		3rd Week	6th Week	9th Week	3rd Week	6th Week	9th Week
Age (Years)	Diet	Median (IQR)	Median (IQR)	Median (IQR)	Median (IQR)	Median (IQR)	Median (IQR)
**<40**	**CD ^a^**	2.3	2.3	3.5	7.8	7.8	0.9
	**n**	1	1	1	1	1	1
	**LGIMD ^b^**	7.8 (1.6; 19.1)	7.8 (1.6; 19.1)	7.8 (1.6; 19.1)	4.4 (0.9; 8.7)	4.4 (0.9; 8.7)	4.4 (0.9; 8.7)
	**n**	8	8	8	8	8	8
**40–49**	**CD ^a^**	4.7 (2.0; 9.4)	4.7 (2.6; 15.2)	4.9 (2.6; 15.2)	2.1 (0.9; 3.3)	2.1 (0.9; 3.3)	2.1 (1.6; 3.3)
	**n**	2	2	2	5	5	4
	**LGIMD ^b^**	3.2 (1.3; 5.6)	3.2 (1.3; 5.6)	3.2 (1.3; 5.6)	2.1 (1.1; 3.2)	2.1 (1.1; 3.2)	2.1 (1.1; 3.2)
	**n**	18	18	18	9	9	9
**≥50**	**CD ^a^**	4.7 (3.4; 12.3)	5.2 (3.4; 12.4)	4.7 (3.7; 12.4)	5.9 (3.6; 9.6)	5.9 (3.6; 9.6)	5.9 (3.6; 9.0)
	**n**	8	7	7	5	3	2
	**LGIMD ^b^**	4.7 (3.4; 12.3)	4.7 (3.4; 12.3)	4.7 (3.4; 12.3)	5.9 (3.7; 9.5)	5.9 (3.7; 9.5)	5.6 (3.3; 9.5)
	**n**	30	30	30	17	17	17

^a^ CD: Control Diet; ^b^ LGIMD: Low Glycemic Index Mediterranean Diet.

**Table 6 nutrients-13-00066-t006:** Repeated Measures Analysis of Variance. Effect of Treatments on NAFLD ^a^ Scores (assessed by CAPb value) by Intervention Arms and Time and Contrast between Intervention and Time. NutriAtt Trial, Castellana Grotte, 2015–2016.

CAP ^b^	β	SE	*p*-Value	(CI 95%)
**Working arms**				
C ^c^	0			
LGIMD ^d^	−55.81	38.61	0.150	(−131.90; 20.29)
PA1 ^e^	−166.35	38.39	0.000	(−242.01; −90.68)
PA2 ^f^	−78.02	38.54	0.044	(−153.97; −2.07)
PA1+LGIMD ^g^	−94.10	38.50	0.014	(−170.87; −19.12)
PA2+LGIMD ^h^	−76.37	38.45	0.048	(−152.14; −0.60)
**Time**				
Baseline	0			
45 days	−11.04	15.28	0.471	(−41.16; 19.07)
90 days	−24.46	16.04	0.129	(−56.07; 7.15)
**Time * Working arms:**				
(45 days vs. base) C ^c^	−11.04	15.28	0.471	(−41.16; 19.07)
(90 days vs. base) C ^c^	−24.46	16.04	0.129	(−56.07; 7.15)
(45 days vs. base) LGIMD ^d^	−44.24	14.10	0.002	(−72.03; −16.45)
(90 days vs. base) LGIMD ^d^	−34.84	14.97	0.021	(−64.34; −5.34)
(45 days vs. base) PA1 ^e^	−37.95	14.10	0.008	(−65.75; −10.14)
(90 days vs. base) PA1 ^e^	−44.50	15.01	0.003	(−74.08; −14.92)
(45 days vs. base) PA2 ^f^	−32.23	13.53	0.018	(−58.88; −5.57)
(90 days vs. base) PA2 ^f^	−46.22	14.61	0.002	(−75.02; −17.43)
(45 days vs. base) PA1+LGIMD ^g^	−52.96	13.23	0.000	(−79.03; −26.88)
(90 days vs. base) PA1+LGIMD ^g^	−61.56	14.23	0.000	(−89.61; −33.50)
(45 days vs. base) PA2+LGIMD ^h^	−34.25	12.95	0.009	(−59.77; −8.73)
(90 days vs. base) PA2+LGIMD ^h^	−38.15	13.38	0.005	(−64.53; −11.77)

^a^ NAFLD: Non-Alcoholic Fatty Liver Disease; ^b^ CAP: Controlled Attenuation Parameter Measured by FibroScan^®^; ^c^ C: Control Diet; ^d^ LGIMD: Low Glycemic Index Mediterranean Diet; ^e^ PA1: Aerobic Activity Program; ^f^ PA2: Combined Activity Program; ^g^ PA1 + LGIMD; ^h^ PA2 + LGIMD.

**Table 7 nutrients-13-00066-t007:** Repeated Measures Analysis of Variance. Effect of Treatments on NAFLD Scores (assessed by CAP value) by, Time and NAFLD Severity. NutriAtt Trial, Castellana Grotte, 2015–2016.

	NAFLD ^a^			
	Moderate		Severe	
CAP ^b^	Contrast	(CI 95%)	Contrast	(CI 95%)
(45 days vs. base) C ^c^	−15.50	(−64.02; 33.02)	−8.73	(−45.28; 27.81)
(90 days vs. base) C ^c^	−29.00	(−77.52; 19.52)	−21.82	(−61.38; 17.73)
(45 days vs. base) LGIMD ^d^	−13.80	(−66.96; 39.36)	−54.27	(−85.89; −22.64) **
(90 days vs. base) LGIMD ^d^	−22.65	(−80.58; 35.28)	−39.13	(−72.53; −5.73) *
(45 days vs. base) PA1 ^e^	−6.67	(−55.19; 41.86)	−51.36	(−84.09; −18.62) *
(90 days vs. base) PA1 ^e^	10.67	(−37.86; 59.19)	−72.77	(−108.69; −36.85) **
(45 days vs. base) PA2 ^f^	18.14	(−26.78; 63.07)	−55.73	(−87.35; −24.11) **
(90 days vs. base) PA2 ^f^	18.40	(−29.24; 66.05)	−77.70	(−112.16; −43.24) **
(45 days vs. base) PA1 + LGIMD ^g^	−23.75	(−65.77; 18.27)	−68.53	(−100.15; −36.91) **
(90 days vs. base) PA1 + LGIMD ^g^	−23.09	(−37.31; 21.13)	−83.27	(−117.72; −48.81) **
(45 days vs. base) PA2 + LGIMD ^h^	−27.4	(−64.99; 10.19)	−39.14	(−71.87; −6.41) *
(90 days vs. base) PA2 + LGIMD ^h^	−46.31	(−85.43; −7.19) *	−32.57	(−66.23; 1.09)

^a^ NAFLD: Non-Alcoholic Fatty Liver Disease; ^b^ CAP: Controlled Attenuation Parameter Measured by FibroScan^®; c^ C: Control Diet; ^d^ LGIMD: Low Glycemic Index Mediterranean Diet; ^e^ PA1: Aerobic Activity Program; ^f^ PA2: Combined Activity Program; ^g^ PA1 + LGIMD; ^h^ PA2 + LGIMD. * *p*-Value < 0.05; ** *p*-Value < 0.001.

## Data Availability

The datasets used in this study are available from the corresponding author.

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
