# Peer review of "Physical Activity and Low Glycemic Index Mediterranean Diet: Main and Modification Effects on NAFLD Score. Results from a Randomized Clinical Trial"

_nutrients, 2020, doi:10.3390/nu13010066_

Round 1

Reviewer 1 Report

The work aims to study the effect of two programs of two different physical activity programs, a Low Glycemic Index Mediterranean Diet (LGIMD), and their combined effect on the he NAFLD score as measured by FibroScan. Six working groups were established. The results obtained in the study show a decrease in the NAFLD score after 45 days of treatment in all groups, except in the control group. The conclusion of this work is that all groups are efficient in reducing NAFLD scores, but Low Glycemic Index Mediterranean Diet plus Aerobic activity program group was the most efficient.

Although the effect of diet, especially the Mediterranean diet, as well as the effect of physical activity on this pathology is known, given the pathology studied, any additional information that can be provided is of scientific interest.

However, there are several aspects that make this work difficult to understand and that should be modified or clarified to give more consistency to the conclusions.

The introduction shows the importance of this pathology and how diet and physical activity can improve it. This information could be improved by giving a little more information about the exercise protocols and the chosen diet. However, the main problem, in my opinion, is that it is not clearly justified why this work is being carried out. That is, what does this study contribute to what is already known about the effect of physical activity and the Mediterranean Diet? This point must be clarified in the text. It remains unclear what makes this study different from other similar ones.

There are two aspects that should be treated in more depth when discussing the results. First the age range, I think it is too wide (30-60 years) and it has been a very little studied factor when drawing conclusions. Second, using people with moderate-range and severe-range NAFLD, surely, the effect of the treatments would not be the same in one case as in another and it has not been studied either.

Although the text indicates that only data related to CAP and Intervention Arms were used in the analysis, I think it would have been interesting to analyze the effect of the rest of the biochemical and anthropometric parameters analyzed. This analysis could help to a better understanding of the observed differences.

Finally, I think it is necessary to indicate that there is a relatively high level of plagiarism, an attempt should be made to reduce it, even if the source is a publication by the authors themselves

Reviewer 2 Report

Table 1 participant variables was too extensive disrupting my flow of reading and I did not know how to use all the data. Can you shorten the list and place the majority in an Appendix or Supplement

In the keywords can you add abbreviations and define CAP

Consider moving the dietary information to an Appendix rather than a Supplement so it would be easier to see directly at the end of this paper. 

Round 2

Reviewer 1 Report

I consider that the article has been modified appropriately.

This manuscript is a resubmission of an earlier submission. The following is a list of the peer review reports and author responses from that submission.

Round 1

Reviewer 1 Report

Very good paper but  I have a few suggestions:

  1. Can you provide a few details about the control diet?
  2. Why does Table 3 contain .(.) under 45 days mean +/- SD?
  3. In the discussion section line 296-301 you list strong adherence as a strength and as a limitation because of lower adherence after 45 days.  

Reviewer 2 Report

It is a potentially interesting work, but tainted by a rather complex
design that limits its conclusions.
Moreover, the number of patients for each subgroup is very limited
in order to reach biologically plausible conclusions.
Another limitation of the study is the duration of the observation
which limits the possible translation of the results into clinical
practice.

Reviewer 3 Report

The first point to consider and why the article should not follow the review process, is the high level of plagiarism. The article has a plagiarism of 45%, that is, almost half of the article. It remains at approximately 35% after eliminating bibliographic references.

THIS PLAGIARISM IS INADMISSIBLE AND IT SHOULD BE THE FIRST THING TO CORRECT.

The objective of the authors in this study is to estimate the effect of two programs of two different physical activity programs, a Low Glycemic Index Mediterranean Diet (LGIMD), and their combined effect on the he NAFLD score as measured by FibroScan.

The authors establish six working groups and, in the conclusions, show that all groups, except the control group, reduce the NAFLD score after 45 days of intervention, with the group that combines LGIMD and an aerobic physical activity program the one with the greatest effect.

The article is interesting given the pathology on which it is working and although the importance of diet, especially the Mediterranean Diet, on this pathology is known, and, at the same time, the effect of physical activity, any additional contribution is always of scientific interest.

However, there are multiple aspects that should be modified.

Introduction

Although the introduction clearly justifies the importance of this pathology and how it can be improved through diet and physical activity, it is not justified in a clear way why this study is carried out. That is, what does this study contribute to what is already known about the effect of physical activity and the Mediterranean Diet? Why have these physical activity protocols been chosen? What are the advantages of a Mediterranean diet with a low glycemic index?

I think it is important to indicate in the introduction what makes this study different from other similar ones.

Material and Methods

There are two aspects that should be treated in more depth when discussing the results. First the age range, I think it is too wide (30-60 years) and it has been a very little studied factor when drawing conclusions. Second, using people with moderate-range and severe-range NAFLD, surely, the effect of the treatments would not be the same in one case as in another and it has not been studied either.Considero que existe un rango de edad excesivamente grande (30-60 años) 166 en material y métodos y 144 en el abstract, rango de edad muy amplio 30-60 años, moderado o severo NAFLD score.

Where is additional table 1 showing Details about the tests, scores and evaluation schedule.

It shows the effect by gender, but is the effect of the physical activity protocols assessed by age? That is, we have people of 30 years and people of 60 years, although the protocols can be adapted in intensity to the age of the person, it is difficult to consider that the effect of the same protocol in a 30-year-old person is the same as in a 60-year-old person.

The material and methods do not indicate the determined blood parameters and how they have been determined.

Results

The gender data are repeated in table 1 and in table 2.

Page 9, line 202: At baseline, more subjects had severe than moderate NAFLD, ¿Only at baseline?

Table 3 lacks data for groups 1 and 2 at 45 days.

Table 4 is incomplete, weeks 1 and 2 are missing, also, 90 days is more than 10 weeks.

Why in table 5, is the "n" not given according to age?

Table 6 is difficult to understand, it should be clarified. It is observed that when considering time, there is a decrease in NAFLD scores in all groups except in the control group, but it is not clear whether when comparing the data in the control group with those observed in other groups for the same time, there are such differences or not. and I think this aspect is important. For example, at 90 days, between groups 1 and 2, I don't know if these significant differences would exist.

Why have the effect on the rest of the anthropometric and biochemical parameters analyzed not been analyzed? This could help to understand the reason for the observed differences and provide more information on the effect of diet and physical activity programs. It does not make sense to give so much data, when finally only one of them is used.

Discussion

Page 13, line 261-265, indicates that after 45 days the effect of the LGIMD group on the NAFLD score decreases because it is difficult to follow healthy habits, although in the data on adherence to the diet it seems that an increase in adherence to the diet over time.

In addition, among the limitations it indicates “the maintenance of the LGIMD, featuring a lower adherence after 45 days”, however, as I have indicated, this does not seem to be shown in the only table that shows adherence to the diet (table 5), except in case for the data shown in women under 40 years of age, which gives some data that should be discussed.

Ultimately, the conclusion shows the strengths of the study, but it should also show the weaknesses and limitations that are, in my opinion, more than one.

Reviewer 4 Report

Here, Franco et al. present a subset of results deriving from a RCT conducted in a population of subjects with NAFLD from Puglia, Italy. N=144 subjects were randomized to receive either control or low-glycemic index diet regimens combined (or not) with two different exercise programmes, one aerobic, and the other mixed aerobic/resistance. Results revealed a significant beneficial effect of all treatments (except control diet) on the primary endpoint, i.e. controlled attenuation parameter measured by Fibroscan.

The trial was well conducted, the sample size was correctly estimated, and therefore results are adequately powered to support the authors' conclusions.

Dietary and excercise programs are described in detail, allowing replication, and adherence to the latter is correctly reported.

Overall, the trial investigates a very important issue, i.e. the rapidly increasing prevalence of metabolic derangements in Italy, especially in the southern Regions. 

I therefore recommend acceptance of the paper, after addressing only a couple of minor comments:

  • I would add in the introduction considerations about the noninvasive assessment of NAFLD through Fibroscan, with some references to previous studies showing the predictive value of CAP towards other endpoints, such as the progression to NASH.
  • Please consider adding in Figure 2 the p-values for post hoc comparisons among groups at each time point, and subsequently update the figure legend to reflect this addition.

Round 2

Reviewer 2 Report

limitations remain on the originality of the work and on the poor translation of the results into clinical practice

Reviewer 3 Report

The text has been improved in various aspects, however, there are still important aspects to consider.

The main problem is still the high existing plagiarism. This point has not been corrected. The author justifies that they are articles published by themselves, but the regulations of the journal, in relation to plagiarism, indicate: “Plagiarism is not acceptable in MDPI journals. Plagiarism includes copying text, ideas, images, or data from another source, even from your own publications, without giving credit to the original source ”. In this case it is not only in the material and methods as indicated by the author, but in many other points, for example, page 15 line 296-298, line 303-306, line 309-316.

AGAIN, THIS PLAGIARISM IS INADMISSIBLE AND IT SHOULD BE THE FIRST THING TO CORRECT.

Introduction

I asked: what does this study contribute to what is already known about the effect of physical activity and the Mediterranean Diet? Why have these physical activity protocols been chosen? What are the advantages of a Mediterranean diet with a low glycemic index?

The authors have clarified the question of the low glycemic index diet and they answer in part to why the activity protocols have been chosen. However, it still does not clarify the question, in my opinion, more importantly, what does this study contribute to what is already known about the effect of physical activity and the Mediterranean Diet?. This point must be clarified in the text, that is, indicate that it is studied in this work that has not been previously studied, since all these points have been studied, but what makes this study different from the rest, or simply re-study already studied. It may be that this diet has never been studied before or these physical activity protocols or the combination of a special diet with physical activity. It remains unclear what makes this study different from other similar ones.

Results

I asked: There are two aspects that should be treated in more depth when discussing the results. First the age range, I think it is too wide (30-60 years) and it has been a very little studied factor when drawing conclusions. Second, using people with moderate-range and severe-range NAFLD, surely, the effect of the treatments would not be the same in one case as in another and it has not been studied either.

The answer given by the authors justifies the age range used (NAFLD is condition arising about 30 years old in both men and women with a peak about 60 years old) but the authors consider that although it is possible that there were differences due to age, these they are not important for the objective pursued. I wonder why the authors believe that these differences are not important. I think that if gender is important to indicate it in a table, so should age. Also, they don't answer my second question: using people with moderate-range and severe-range NAFLD, surely, the effect of the treatments would not be the same in one case as in another and it has not been studied either. I think it should be explained if differences have been observed between those who have a moderate degree or those who have a severe degree.

Results

Why in table 5, is the "n" not given according to age?

The "n" has been corrected, but the data in the table does not make sense. What I would like to know is how many men under 40 are there? and women? And between 40-49 years? Etc. The "n" has been given based on gender (table 1), but not based on age
